# Probabilistic Forecasting of Short-Term Electric Load Demand: An Integration Scheme Based on Correlation Analysis and Improved Weighted Extreme Learning Machine

**Zhengmin Kong** , **Zhou Xia \*** **and Yande Cui** **and He Lv**

School of Electrical Engineering and Automation; Wuhan University, Wuhan 430072, China; zmkong@whu.edu.cn (Z.K.); cuiyande@whu.edu.cn (Y.C.); oubahe@126.com (H.L.)

\* Correspondence: summerboat@whu.edu.cn; Tel.: +86-182-0710-7791

**Abstract:** Precise prediction of short-term electric load demand is the key for developing power market strategies. Due to the dynamic environment of short-term load forecasting, probabilistic forecasting has become the center of attention for its ability of representing uncertainty. In this paper, an integration scheme mainly composed of correlation analysis and improved weighted extreme learning machine is proposed for probabilistic load forecasting. In this scheme, a novel cooperation of wavelet packet transform and correlation analysis is developed to deal with the data noise. Meanwhile, an improved weighted extreme learning machine with a new switch algorithm is provided to effectively obtain stable forecasting results. The probabilistic forecasting task is then accomplished by generating the confidence intervals with the Gaussian process. The proposed integration scheme, tested by actual data from Global Energy Forecasting Competition, is proved to have a better performance in graphic and numerical results than the other available methods.

**Keywords:** probabilistic forecasting; integration scheme; correlation analysis; improved weighted extreme learning machine; switch algorithm; Gaussian process

---

## 1. Introduction

Electric load forecasting (LF) is an indispensable task for electric utilities in the long term [1]. As a fundamental business problem, LF is associated with decision-making processes in power system plannings, operations, energy trading and so forth [2,3]. Prediction of future load demand is a tendency forecasting subject based on actual load data. Methods of tendency forecasting are generally classified into three categories: the physical approaches, the statistical approaches and the artificial intelligence (AI) approaches [4]. In practice, the diversity of data sources makes it impossible to build physical models for LF [5]. Moreover, since the load demand is affected by the chaotic nature of weather conditions, statistical approaches such as exponential smoothing, auto regression (AR), moving average (MA) and their variants are insufficient to address the nonlinearity and randomness properties of load data [6]. Consequently, AI approaches have become the mainstream of LF methods. Traditional LF studies apply AI-based methods of neural network [7–9], support vector machine [10–12] and other methods [13,14] to forecast the load demand with exactly values. Although the performance of traditional LF models can be improved by optimizing the AI model, the forecasting uncertainty at each load point is unknown [15]. To develop efficient strategies for power market management, more detail about forecasting results should be provided [16].

Preferably, an emerging technique called probabilistic forecasting can offer much more comprehensive information about future tendency, and thus is more effective for decision-making

in the dynamic environment [17]. Over the past few years, numerous probabilistic forecasting methods have been proposed and employed to achieve probabilistic forecasts in the industrial tendency analysis [18]. Generally, more steps are involved in generating probabilistic intervals than point forecasting. Hence, gradient-based methods with long training time are improper for probabilistic forecasting [19,20]. In the field of LF, common forms of probabilistic load forecasting (PLF) refer to providing output with quantile, interval and density function [21]. It is noticed that the interval methods include prediction interval (PI) and confidence interval (CI).

Despite the investigations of PLF published in Global Energy Forecasting Competition 2014 (GEFCom2014) [22–24], advanced techniques for PLF are quite limited. In addition, most of the investigations focus on the quantile and interval methods. Typical research on introducing the quantile regression averaging (QRA) to PLF can be found in [21], which employed the QRA technique in point forecasts for PLF tasks. In addition, an advanced quantile regression technique called gradient boosting machine (GBM) is proved to outperform QRA techniques in terms of robustness and accessibility [25]. However, a large amount of time in training 99 required quantiles is unpalatable for short-term load demand forecasting. The emerging technique of quantile regression neural network (QRNN) seems to be a cure for reducing the time cost by optimizing a single forecasting model with overall pinball loss [26]. Nevertheless, the massive computation brought by a large data set is unsustainable. The PI method mentioned in [27] estimated the variance of forecasting errors with load forecasts based on weather ensembles. In [28], the generalized extreme learning machine (ELM) was developed to predict the load interval. Both the prediction error and the noise uncertainty were taken into account in this ELM. Since the PIs are obtained by prediction, the accuracy of PI methods is only guaranteed for fixed interval size. Moreover, the two-stage neural network [29] and the bootstrapping technique [30] provide a statistical approach to generate CIs, of which the interval size can be adjusted flexibly with various confidence levels. The width of CIs highly depends on the behavior of forecasting process. Thus, the CI methods are sensitive to the quality of load data, namely the noise factors might result in redundant uncertainty due to the fluctuations in interval size.

However, for the short-term load forecasting (STLF) in an hour-ahead situation, the procedure of LF must be effective so that market operators and participants can react in time [31]. Hence, the massive computation and time cost of quantile methods are unbearable. Although interval methods with the application of ELM can overcome these limitations by removing training steps, the accuracy of output highly depends on its random given parameters [32]. In addition, the fixed intervals of PI methods and the redundant uncertainty of CI methods are adverse. Meanwhile, the impacts of load data noise on the PLF can not be neglected [33]. Variable selection and feature extraction are the critical steps for better performance in PLF. Xie [34] and Reis [35] summarized the frequently-used tools in load data processing, among which correlation analysis and wavelet transform were the most typical and efficient ones to be integrated into load forecasting ensembles [36].

In this paper, an integration scheme for PLF is proposed in order to strike a trade-off among the advanced techniques mentioned above. The proposed scheme is a CI method that integrates the techniques of wavelet packet transform, correlation analysis, Gaussian process and improved weighted ELM. More specifically, the load data in time series form are selected and extracted by the cooperation of wavelet packet transform and correlation analysis. The improved weighted ELM with a computation reducing switch algorithm is the core of the forecasting model, while the Gaussian process provides probabilistic intervals. Furthermore, the load data are separated into principal parts and noise. The principal parts are utilized to load forecasting, while the noise, as a part of total forecasting uncertainty, is merged with the model uncertainty.

The contributions of this paper are summarized as follows:

(1) The innovative scheme of combining wavelet packet transform and correlation analysis methods is proposed to remove noise from raw data.
(2) The improved weighted ELM is applied for forecasting, and a computation reducing switch algorithm is presented to obtain stable parameters for this ELM.

(3)  Considering the forecasting uncertainty, the Gaussian process is creatively used to deal with the noise uncertainty and generate CIs for PLF.
(4)  Newly raised criteria and an indicator are adopted for efficiency validation and contrast between different PLF methods involved in comparisons.

The remainder of the paper is organized as follows. In Section 2, the techniques involved in the integration scheme are elaborated sequentially. Section 3 explains the whole procedure for implementing load forecasting in detail. Validation and case study with real data are demonstrated in Section 4. Section 5 provides analysis and discussion of the results, while Section 6 concludes the whole paper.

## 2. Methodology

### 2.1. Wavelet Packet Transform (WPT)

Wavelet transform (WT) is an efficient tool for signal analysis in the time-frequency domain. As an extension of WT, WPT further decomposes the detailed information in the high-frequency region [37]. This ability makes it possible for WPT to be applied to remove Gaussian noise from the original signal. Based on the multi-resolution analysis (MRA) [38], the procedure for a 3-level signal decomposition is shown in Figure 1. The load series S is broken down into approximation (A) components and detail (D) components.

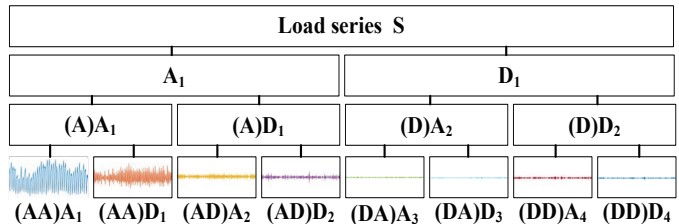

**Figure 1.** Procedure for signal decomposition using wavelet packet transform (WPT).

Most of the load data in STLF have complex detail components due to the volatile environment. A more precise forecast may be available after neglecting these components [39]. The wavelet bases of Daubechies (db) and Coiflets (coif) are proved to be capable of treating the load data [40]. In this paper, wavelet families of db2-db5 and coif2-coif5 are involved in data processing.

### 2.2. Correlation Analysis

As a basic method in statistics, correlation analysis is used to dig out the relationship between a pair of variables. For predicting future behavior, it is essential that the variables have the similar trend or distribution. An expression called the Pearson correlation coefficient (PCC) is commonly recognized as the standard of correlation analysis. The value of PCC ranges from $-1$ to 1, with $-1$, 0, 1 indicating that the two variables are perfectly negatively correlated, uncorrelated and perfect positive correlated, respectively. An autocorrelation coefficient (AC) is a special case of PCC where the variables are the sub-series of the same series with various time series indexes. By defining $S(t_0)$ and $S(t_1)$ as the sub-series of variable $S$ with time series indexes $t_0$ and $t_i$ ($i \neq 0$), the AC is defined as

$$\rho_{S(t_i)} = \frac{Cov\left(S(t_0), S(t_i)\right)}{\sqrt{D\left(S(t_0)\right)}\sqrt{D\left(S(t_i)\right)}} = \frac{Cov\left(S(t_0), S(t_i)\right)}{D\left(S\right)}, \tag{1}$$

where $\rho_{S(t_i)}$ is the AC between variables $S(t_0)$ and $S(t_1)$, $Cov\left(\cdot\right)$ is the covariance, and $D\left(\cdot\right)$ is the variance. $\rho_{S(t_i)}$ is determined by $t_i$ only. For both PCC and AC, it is generally accepted that a little correlation exists when their absolute values are below 0.3 and the absolute values exceed 0.5 means that the series are related [41].

In this paper, the two above standards are adopted to pick out the relevant sub-series from load data and extract features from the wavelet components, which have stronger autocorrelation than the origin sub-series. The proposed method is shown in Figure 2.

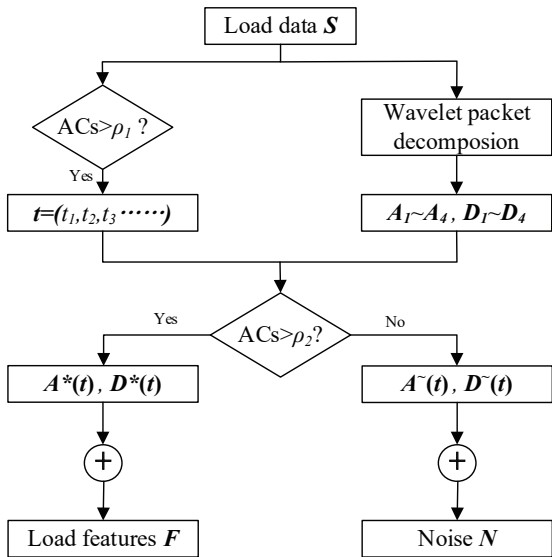

**Figure 2.** Flowchart of the correction analysis method.

According to Figure 2, the load data $S$ is divided into load features $F$ and noise $N$. $\rho_1$ and $\rho_2$ are the standards for load data filtering in time domain and wavelet domain. $A^*(t), D^*(t)$ and $A^\sim(t), D^\sim(t)$ are the sub-series of wavelet components at time series $t$. To be proportional to correlation, ACs are in the form of absolute value. Indeed, the PCCs between different approximation and detail components are near zero since the components are decomposed by orthogonal wavelet bases. Then, Equation (1) is also expressed as

$$
\begin{aligned}
\rho_{S(t_i)} &= \frac{Cov\left(S(t_0), S(t_i)\right)}{\sqrt{D\left(S(t_0)\right)}\sqrt{D\left(S(t_i)\right)}} = \frac{Cov\left(F(t_0)+N(t_0), F(t_i)+N(t_i)\right)}{\sqrt{D\left(F(t_0)+N(t_0)\right)}\sqrt{D\left(F(t_i)+N(t_i)\right)}} \\
&= \frac{Cov\left(F(t_0),F(t_i)\right)+Cov\left(F(t_0),N(t_i)\right)+Cov\left(N(t_0),F(t_i)\right)+Cov\left(N(t_0),N(t_i)\right)}{\sqrt{D\left(F(t_0)\right)+D\left(N(t_0)\right)+2Cov\left(F(t_0),N(t_0)\right)}\cdot\sqrt{D\left(F(t_i)\right)+D\left(N(t_i)\right)+2Cov\left(F(t_i),N(t_i)\right)}} \\
&\approx \frac{Cov\left(F(t_0),F(t_i)\right)+Cov\left(N(t_0),N(t_i)\right)}{D\left(F(t_0)\right)+D\left(N(t_0)\right)}.
\end{aligned}
\tag{2}
$$

Setting $\rho_1 \geq \rho_2$, the relation for ACs of $S$ and $N$ is obtained as

$$
\rho_{S(t_i)} > \rho_1 \geq \rho_2 > \rho_{N(t_i)}, \left(t_i \in t\right).
\tag{3}
$$

Combining Equation (2) and Equation (3), the connection for ACs of $S$ and $F$ can be formulated as follows:

$$
\begin{aligned}
\rho_{F(t_i)} - \rho_{S(t_i)} &= \frac{Cov\left(F(t_0),F(t_i)\right)}{D\left(F(t_0)\right)} - \frac{Cov\left(F(t_0),F(t_i)\right)+Cov\left(N(t_0),N(t_i)\right)}{D\left(F(t_0)\right)+D\left(N(t_0)\right)} \\
&= \frac{Cov\left(F(t_0),F(t_i)\right)D\left(N(t_0)\right)}{D\left(F(t_0)\right)\cdot\left[D\left(F(t_0)\right)+D\left(N(t_0)\right)\right]} - \frac{Cov\left(N(t_0),N(t_i)\right)D\left(F(t_0)\right)}{D\left(F(t_0)\right)\cdot\left[D\left(F(t_0)\right)+D\left(N(t_0)\right)\right]} \\
&= \frac{D\left(N(t_0)\right)}{D\left(F(t_0)\right)}\cdot\left(\rho_{S(t_i)}-\rho_{N(t_i)}\right) > 0.
\end{aligned}
\tag{4}
$$

Based on the analysis above, it is concluded that $F$ has stronger autocorrelation than $S$ and is better for tendency forecasting. $N$ is a series with little autocorrelation which contributes nearly nothing to

future prediction. The distribution of *N* can be approximated by Gaussian distribution. Extracted from actual load series, distributions of *N* can be seen in Figure 3.

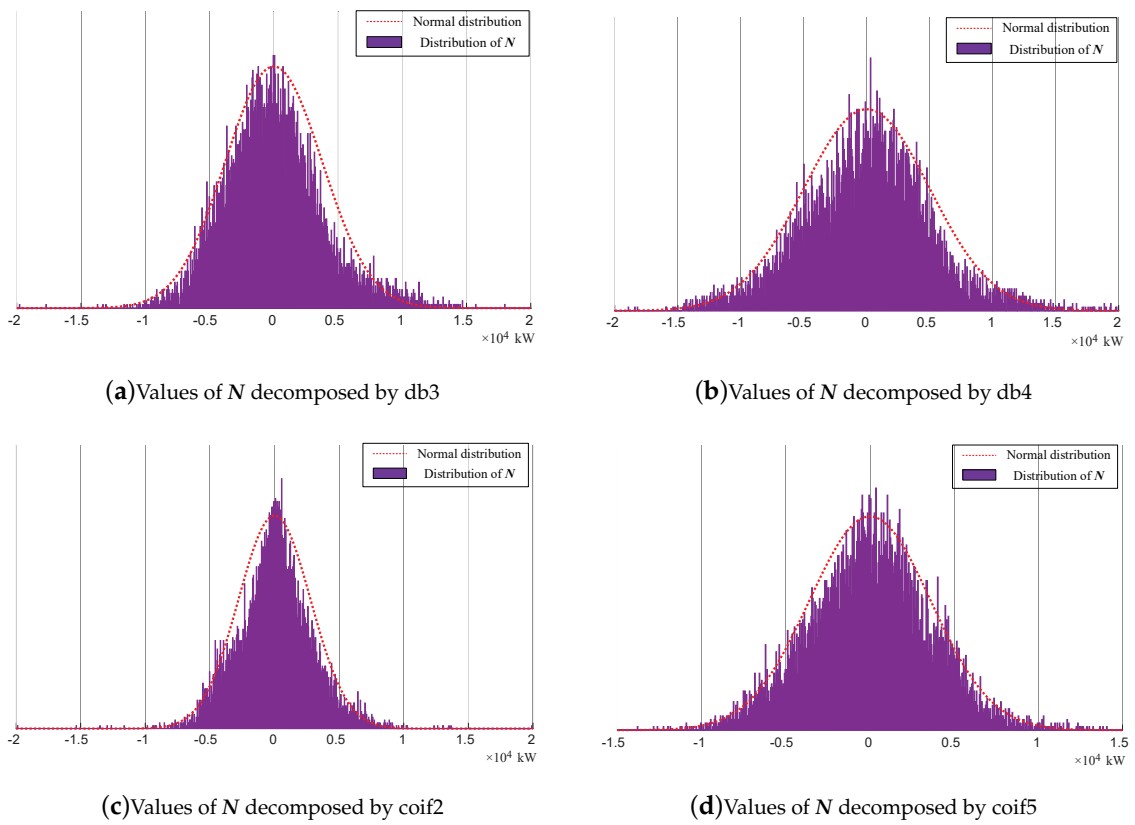

**Figure 3.** The distributions of *N* compare to normal distribution.

*2.3. Gaussian Process*

Gaussian (Normal) distribution is the most common continuous probability distribution in statistics. The notation of a Gaussian distribution with mean value $\mu$ and variance $\sigma^2$ is represented as $N \sim (\mu, \sigma^2)$. Its probability density function(PDF) is defined as

$$f(x) = \frac{1}{\sqrt{2\pi}\sigma} e^{\left(-\frac{(x-\mu)^2}{2\sigma^2}\right)},\tag{5}$$

where $x$ is a random variable. Preferably, for $d$ independent univariate Gaussian distributions $N_\alpha \sim (\mu_\alpha, \sigma_\alpha^2), \alpha = 1, 2, \cdots\cdots d$, it is well-known the product and convolution of their PDFs are also Gaussian PDFs [42]. Define $N_p \sim (\mu_p, \sigma_p^2)$ and $N_c \sim (\mu_c, \sigma_c^2)$ as the new Gaussian distribution for the product and convolution of $d$ independent univariate Gaussian distributions. It can be calculated that

$$\mu_p = \sum_{\alpha=1}^{d} \left(\frac{\mu_\alpha}{\sigma_\alpha^2} \cdot\right) \sigma_p^2 \qquad \sigma_p^2 = \sum_{\alpha=1}^{d} \frac{1}{\sigma_\alpha^2},\tag{6}$$

$$\mu_c = \sum_{\alpha=1}^{d} \mu_\alpha \qquad \sigma_c^2 = \sum_{\alpha=1}^{d} \sigma_\alpha^2,\tag{7}$$

where the $\mu_p, \sigma_p^2$ are the mean value and variance for the product of Gaussian PDFs, and $\mu_c, \sigma_c^2$ are for the convolution case. Normally, the product process mainly works on combining the independent

estimations of the Gaussian distribution, while the convolution process accumulates the uncertainty caused by independent factors.

*2.4. Improved Weighted Extreme Learning Machine (IWELM)*

(1) Basic ELM: ELM is a single layer feed-forward network which is widely accepted for its simple training process and excellent performance in generalization. An origin ELM model on one of $N$ samples $(\mathbf{x}_n, r_n), n = 1, 2, \cdots\cdots N$ can be described by

$$\sum_{l=1}^{L} g\left(\mathbf{x}_n \cdot \mathbf{w}_l + b_l\right)\beta_l = o_n, \tag{8}$$

where $\boldsymbol{x_n} = [x_{n1}, x_{n2}\ldots\ldots x_{nK}]$ is the input vector, $r_n$ and $o_n$ are desired and actual output, respectively. $\boldsymbol{w_l} = [w_{1l}, w_{2l}\ldots\ldots w_{Kl}]^T, l = 1, 2, \cdots\cdots L$ is the input weight, $b_l$ is the bias of hidden layer, and $\beta_l$ is the output weight. For all training samples, the whole model can be written as

$$\mathbf{H}\boldsymbol{\beta} = g\left(\mathbf{X} \cdot \mathbf{W} + \boldsymbol{u} \cdot \boldsymbol{b}\right)\boldsymbol{\beta} = \boldsymbol{o}, \tag{9}$$

where $\mathbf{X} = \left[\boldsymbol{x_1}^T, \boldsymbol{x_2}^T\ldots\ldots \boldsymbol{x_n}^T\right]^T$ is the input matrix, $\mathbf{W} = \left[\boldsymbol{w_1}^T, \boldsymbol{w_2}^T\ldots\ldots \boldsymbol{w_L}^T\right]^T$ is the input weight matrix, $\boldsymbol{u}$ is a N-dimensional column vector in which all the elements are 1, $\boldsymbol{b} = [b_1, b_2\ldots\ldots b_L]$ is the bias vector, $\boldsymbol{r} = [r_1, r_2\ldots\ldots r_N]^T$ is the output vector, and $\mathbf{H}$ is the hidden layer output matrix, which is obtained by giving $\mathbf{W}, \boldsymbol{b}$ randomly. Then, the output weight vector can be directly calculated by $\boldsymbol{\beta} = \mathbf{H}^{\dagger}\boldsymbol{r}$, where $\mathbf{H}^{\dagger}$ is the Moore–Penrose generalized inverse of $\mathbf{H}$.

(2) Weighted ELM: Considering the importance degree of features, a branch of ELM called weighted extreme learning machine (WELM) inserts a coefficient matrix $\mathbf{Y} = \text{diag}\left[\gamma_1, \gamma_2\ldots\ldots \gamma_K\right]$ between $\mathbf{X}$ and $\mathbf{W}$. WELM associates each input feature with the hidden layer through a controllable parameter $\gamma_k \in \mathbf{Y}\left(k = 1, 2\cdots\cdots K\right)$. The whole model is expressed by rewriting Equation (9) as

$$\mathbf{H_Y}\boldsymbol{\beta} = g\left(\mathbf{X} \cdot \mathbf{Y} \cdot \mathbf{W} + \boldsymbol{u} \cdot \boldsymbol{b}\right)\boldsymbol{\beta} = \boldsymbol{o}. \tag{10}$$

For all the training data, the effect of each input feature on hidden layer nodes can be adjusted flexibly as is shown in Figure 4. Overall, $\mathbf{Y}$ and $\boldsymbol{\beta}$ are respectively regarded as the input and output structural parameters of the WELM model.

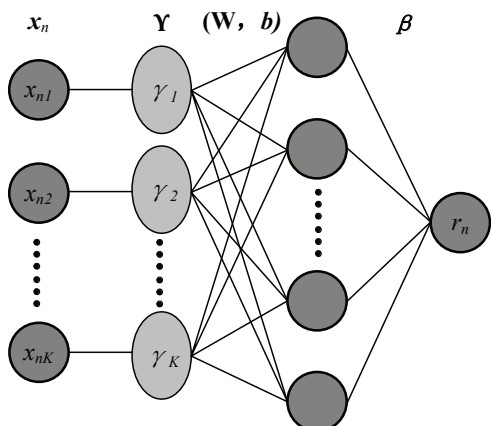

**Figure 4.** The structure of the weighted extreme learning machine (WELM) model.

However, there are two problems with the application of WELM: on the one hand, compared with the increasing computation brought by extra parameters $\gamma_k$, the significance of these extra parameters to LF is ambiguous. On the other hand, the results of this model are various owing to its random

weights and biases. In this paper, an improved WELM model with an optimization algorithm is provided to resolve the above problems.

(3) IWELM: Firstly, the coefficient matrix **Y** is divided into blocks so as to tackle the redundancies brought by the similarity of different features. The blocks of the input features decomposed by a certain wavelet family is proved to be capable of forecasting future load demand as an individual predictor [43]. To take advantage of all the individual predictors, the structure of these predictors is specified by **Y**. For example, for all the features $x_{nk}$ ($k \in Z, k \in [1, K]$) decomposed by db2, it is defined that $\gamma_k = \gamma_{db2}$. $\gamma_{db2}$ is the weight of the above features in the hidden layer nodes and the output.

Secondly, instead of picking the parameters randomly, an improved switch optimization algorithm is presented to attain fixed values for **Y** and **W**, *b*. For minimizing the forecasting errors, the loss function $\eta$ of training data are defined as

$$\eta = \|\mathbf{H}\boldsymbol{\beta} - \mathbf{r}\|^2 = \sum_{i=1}^{N} \xi_i^2 = \sum_{i=1}^{N} (o_i - r_i)^2, \tag{11}$$

where $\|\cdot\|$ stands for the $L^2$ vector norm. $\xi_i$ is the output error of the $i^{th}$ sample. To avoid overfitting in optimization, the Tikhonov regularization [44] is used to restrict the norm of output weights while minimizing the loss. Then, the loss function is rewritten as

$$\eta_G = C\|\mathbf{H}\boldsymbol{\beta} - \mathbf{r}\|^2 + \|\boldsymbol{\beta}\|^2, \tag{12}$$

where $\eta_G$ is the generalized loss function, and $C$ is the regularization parameter. Using the Karush-Kuhn-Tucker (KKT) conditions [45], the optimal equation for $\boldsymbol{\beta}$ is solved as

$$\frac{\partial \eta_G}{\partial \boldsymbol{\beta}} = 0 \rightarrow \boldsymbol{\beta} = \left(\frac{\mathbf{I}}{C} + \mathbf{H}^T \mathbf{H}\right)^{-1} \mathbf{H}^T \mathbf{r}, \tag{13}$$

where $\frac{\partial \cdot}{\partial \cdot}$ is the symbol of partial derivative. After acquiring $\boldsymbol{\beta}$, the minimization of $\eta_G$ tends to be a optimal problem with **Y** and **W**, *b*, which is usually solved by a gradient descent (GD) iterative method.

Iterative algorithms seldom occur in ELM networks due to the massive computation of calculating the generalized inverse matrix in the iterative loops. An advanced algorithm named Levenberg–Marquardt (LM) is applicable for ELM [46], but the number of parameters is strictly limited.

In this paper, an improved switch optimization algorithm based on the special structure of IWELM is proposed. Defining $\dot{H}$, $\dot{\boldsymbol{\beta}}$ and $\dot{r}$ as the targets of optimization and output, the target model is expressed by

$$\dot{r} = \dot{H}\dot{\boldsymbol{\beta}} = (\mathbf{H} + \Delta\mathbf{H})(\boldsymbol{\beta} + \Delta\boldsymbol{\beta}). \tag{14}$$

Then, the switch algorithm is employed to eliminate $\Delta\mathbf{H}$, $\Delta\boldsymbol{\beta}$, which are the redundant parts of **H** and $\boldsymbol{\beta}$, respectively. The scheme for the proposed switch algorithm is shown in Figure 5. This switch algorithm reserves the accuracy of the iterative algorithm and reduces the computation through a 2-layer loops frame that is composed of the LM and Adam algorithm. A specific criterion is involved in switching from LM to Adam.

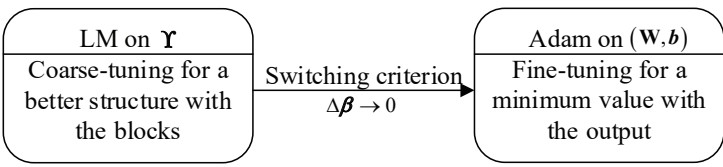

**Figure 5.** The principle for the proposed switch algorithm.

The coarse-tuning is a rapid adjustment on structural parameters $\mathbf{Y}$ and $\boldsymbol{\beta}$, where $\mathbf{Y}$ is optimized by the LM algorithm and $\boldsymbol{\beta}$ is calculated by Equation (13). The fine-turning is an iterative process which aims to obtain accurate values for $\mathbf{W}$ and $\boldsymbol{b}$. The steps of the proposed switch algorithm are outlined in Algorithm 1. The symbol $\otimes$ means the element-wise multiplication. $\boldsymbol{J}$ indicates the Jacobian matrix, and $\kappa$ represents for the switching criterion. Further details about the original switch algorithm can be consulted in [47].

According to the steps in Algorithm 1, the first period of each outer loop is the LM algorithm that is a trust-region algorithm. On this account, the proposed switch algorithm is convergent. The ultimate objective of the switch algorithm is a certain IWELM model with stable parameters $\mathbf{Y}$, $\mathbf{W}$, $\boldsymbol{b}$, $\boldsymbol{\beta}$. The computation is reduced by removing the calculation of the generalized inverse matrix from the iterative process.

---

**Algorithm 1** Switch from LM to Adam.

---

1: **Initialize:** $\delta = 10^{-6}$, $\beta_1 = 0.9$, $\beta_2 = 0.999$, $T_2 = 20$, $\mathbf{Y} = \mathbf{IW}$ and $b$ are given randomly $\boldsymbol{\beta} = \mathbf{H}^\dagger r$, $\eta = \|\mathbf{H}\boldsymbol{\beta} - \mathbf{r}\|^2$

2: **Begin:**

3: **while** $\eta_0 < \eta$ **do**

4: 　　$\kappa = 1$, $t_1 = 0$

5: 　　**while** $\kappa > 10^{-3}$ & $t_1 < T_1$ **do** 　　(LM)

6: 　　　　$\mathbf{J} = \left[ \left(\frac{\partial \xi_1}{\partial \mathbf{Y}}\right)^T, \left(\frac{\partial \xi_2}{\partial \mathbf{Y}}\right)^T \ldots\ldots \left(\frac{\partial \xi_n}{\partial \mathbf{Y}}\right)^T \right]^T$, $\quad \Delta \mathbf{Y} = - \left(\mathbf{J}^T \mathbf{J} + \lambda I\right) \mathbf{J}^T \left(H\boldsymbol{\beta} - r\right)$

7: 　　　　$\mathbf{Y}' = \mathbf{Y} + \Delta \mathbf{Y}$, $\quad \boldsymbol{\beta}' = \left(I/C + \mathbf{H}^T \mathbf{H}\right)^{-1} \mathbf{H}^T r$

8: 　　　　**if** $\eta < \|\mathbf{H}\boldsymbol{\beta}' - r\|$ **then**

9: 　　　　　　$\kappa = \|\boldsymbol{\beta}' - \boldsymbol{\beta}\| / \|\boldsymbol{\beta}\|$, $\quad \eta \leftarrow \|\mathbf{H}\boldsymbol{\beta}' - r\|$

10: 　　　　　　$\boldsymbol{\beta} \leftarrow \boldsymbol{\beta}'$, $\quad \mathbf{Y} \leftarrow'$, $\quad \lambda \leftarrow \lambda/10$

11: 　　　　**else**

12: 　　　　　　$\lambda \leftarrow 10\lambda$, $\quad t_1 \leftarrow t_1 + 1$

13: 　　　　**end if**

14: 　　**end while**

15: 　　$t_2 = 0$, $\quad \boldsymbol{u}_0 = 0$, $\quad \boldsymbol{v}_0 = 0$

16: 　　**while** $t_2 < T_2$ **do** 　　(Adam)

17: 　　　　$\boldsymbol{g}_{t_2} = \partial \eta / \partial \left(\mathbf{W}, \boldsymbol{b}\right)$, $\quad t_2 \leftarrow t_2 + 1$, $\quad \boldsymbol{u}_{t_2} = \frac{\beta_1}{1 - \beta_1^{t_2}} \boldsymbol{u}_{t_2 - 1} + \frac{1 - \beta_1}{1 - \beta_1^{t_2}} \boldsymbol{g}_{t_2}$

18: 　　　　$\boldsymbol{v}_{t_2} = \frac{\beta_2}{1 - \beta_2^{t_2}} \boldsymbol{v}_{t_2 - 1} + \frac{1 - \beta_2}{1 - \beta_2^{t_2}} \boldsymbol{g}_{t_2} \otimes \boldsymbol{g}_{t_2}$, $\quad \left(\mathbf{W}, \boldsymbol{b}\right) \leftarrow \left(\mathbf{W}, \boldsymbol{b}\right) + 0.001 \cdot \frac{\boldsymbol{u}_{t_2}}{\sqrt{\boldsymbol{v}_{t_2}} + \varepsilon}$

19: 　　**end while**

20: **end while**

21: **Return** 　$\mathbf{Y}, \mathbf{W}, \boldsymbol{b}, \boldsymbol{\beta}$

---

## 3. Implementation

In practice, the load data set is divided into training samples and testing samples for validation. To accomplish the PLF task, an integration scheme composed of the methods in Section 2 is proposed. The flowchart of the integration scheme is shown in Figure 6. As is seen in the flowchart, training samples are broken down into components by the method mentioned in Figure 2, and testing samples are extracted with the same indexes in the time domain and wavelet domain. Then, the Gaussian method in Equation (6) is employed to obtain the Gaussian distribution $N \sim \left(\mu_r, \sigma_r^2\right)$ from the homogeneous components decomposed by the wavelet families, where $\mu_r$ is treated as $\dot{r}$ in IWELM model and $\sigma_r^2$ is the variance of Gaussian noise. An example for this Gaussian process can be seen in Figure 7.

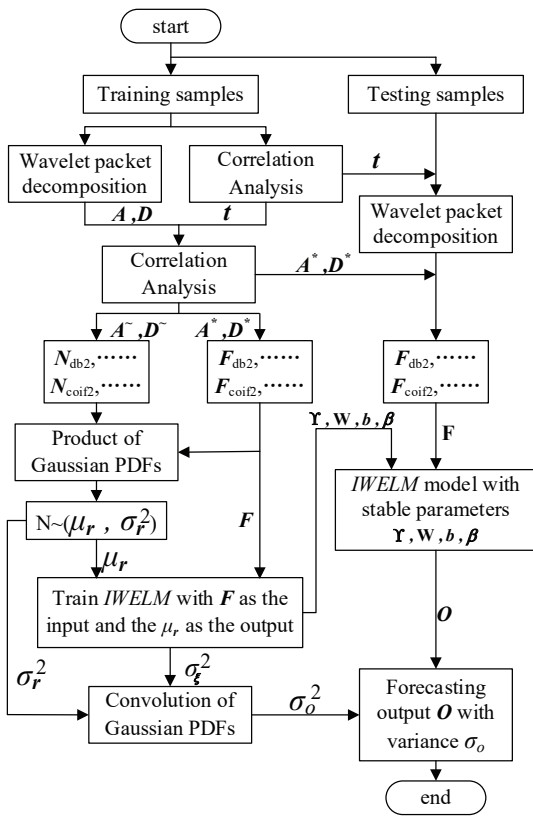

**Figure 6.** Flowchart of the proposed integration scheme.

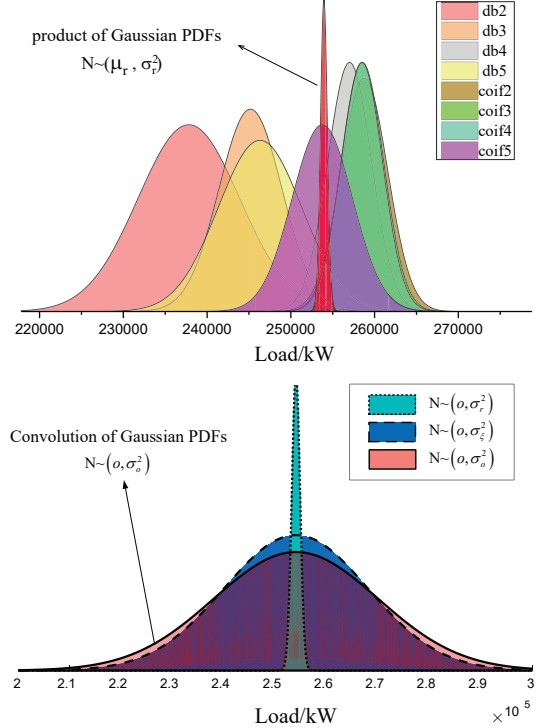

**Figure 7.** The product and convolution of Gaussian probability density functions (PDFs) on an exact load value.

In the next step, all features $F$ decomposed by the involved wavelet family are gathered as the input of the IWELM model. Then, the model is trained with Algorithm 1 on training samples $(F, \dot{r})$. $\sigma_\xi^2$ is the variance of errors between $\dot{r}$ and training output, which is also deemed as the model uncertainty.

Finally, based on the testing samples, the trained IWELM model is exploited to obtain the forecasting output $o$. The total uncertainty $\sigma_o^2$ is the convolution of the $\sigma_r^2$ and $\sigma_\xi^2$, and the forecasting output of the whole scheme is $N \sim (o, \sigma_o^2)$.

## 4. Results

### 4.1. Data Sets

The load data are derived from the files released in GEFcom2014 [22]. Both load data and temperature data are provided in these files. Although additional input features of temperature data and other exogenous variables are helpful to STLF in some cases, the redundant part of these features will add into uncertainties for PLF [23,28,40]. In this paper, hourly load data of four stations are picked out as the forecasting data sets. The load data of these areas differ in orders of magnitude. For each station, load data of 14 months are involved. The load data of 2005 are collected as the training samples, while the first 2 months of 2006 are separated as the testing samples. The ranges of data from stations 1–4 are, respectively, [80,290], [3,18], [10,60], [49,215] $\times 10^3$ (kW). The standardization step in features processing is skipped since only load data are used. The load data part of the data sets is shown in Figure 8.

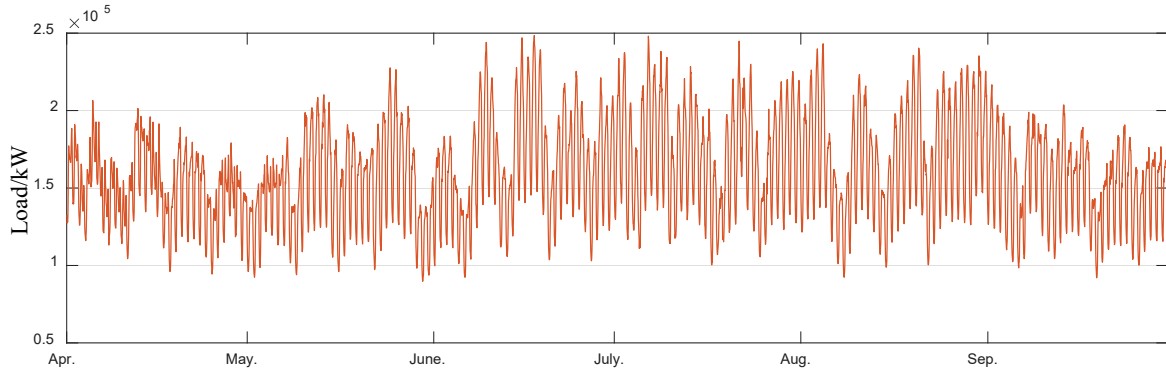

**Figure 8.** Load data of the second and third quarters in 2005.

### 4.2. Models for Comparison

To confirm the superiority of the proposed integration scheme, three state-of-the-art methods of PLF are adopted as comparisons, which are successively quantile regression, prediction interval and confidence interval methods.

**Model 1** (*QRNN*): The quantile regression neural network (QRNN) model discussed in [48] is directly used for PLF. The Huber function [47] is the substitute of the loss function, which is a hybrid of the absolute value and squared error functions.

**Model 2** (*GELM+NN*): The statement of this model can be found in [28]. A wavelet processing with Daubechies is applied to the generalized ELM for load forecasting. The total uncertainty for PLF is the summation of model uncertainty and noise uncertainty. The noise is predicted by a neural network.

**Model 3** (*hybrid ELM+Bootstrapping*): This model is an extension of the load forecasting model published in [40]. The hybrid neural network with an ensemble feature selection method is used as the main part of load forecasting, while the bootstrapping method works for PLF. This model is also regarded as the reference of the proposed model that neglects the noise factor.

### 4.3. Evaluation Criteria and Result

Validating the proposed model with load data of station 1, the forecasting results are formed from load values with CIs that are determined by the total uncertainty. The former and latter 200 h of forecasting results are respectively shown in Figures 9 and 10. The palettes attached to the right of the figures indicate that the confidence level ranges from 0% to 100%, which corresponds to the color changes from dark green to light green. It can be recognized that the results are credible most of the time. Table 1 presents the accuracy of the forecasting results under various confidence levels. The confidence level can be obtained by setting the value of $z$ according to the standard normal ($z$) distribution table. Then, the accuracy is represented by the coverage of CIs ranging from $\mu - z\sigma$ to $\mu + z\sigma$.

It is evident that the larger interval size can improve the forecasting accuracy, though the additional uncertainty is undesirable. For the purpose of a pertinent contrast between different classifications in PLF methods, three criteria are designed here to reach a fair comparison among the particular ones:

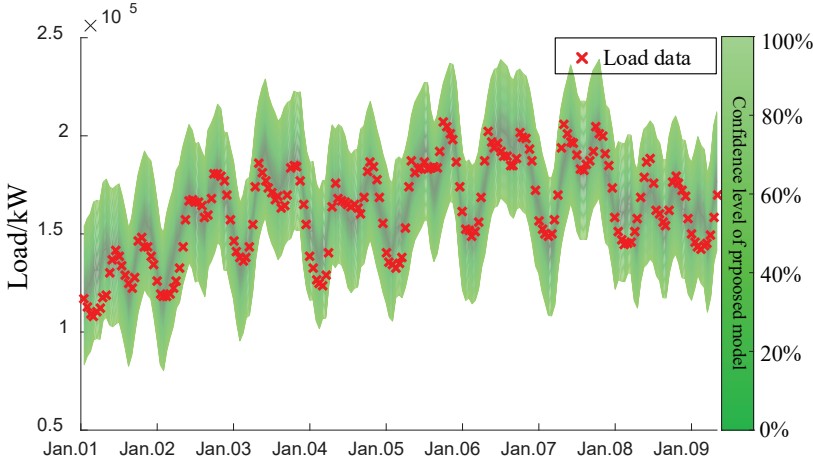

**Figure 9.** The former 200 h of results predicted by the proposed model.

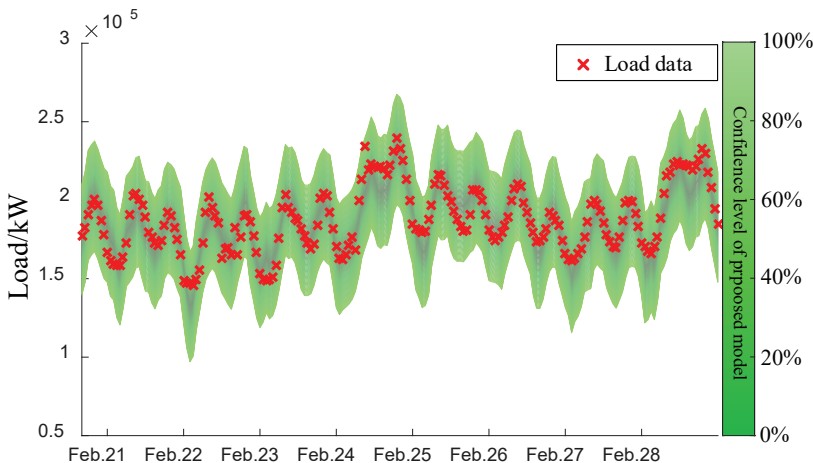

**Figure 10.** The latter 200 h of results predicted by the proposed model.

*(1) Accuracy under the same confidence level* (A/CL): A/CL is obtained by calculating the accuracy under an exact quantile or confidence level. This criterion is suitable for the comparison of quantile regression and CI methods, both of which aim at variable interval size. The graphical results of Model 1 and proposed model under quantile and confidence level of 50% are displayed in Figure 11. As shown in Table 1, the Model 1 performs better than the proposed model only when

the quantile/confidence-level is set to be low, i.e., 10% and 20%. As the quantile/confidence-level increases, the proposed model is more accurate than Model 1 and it better adapts to the given quantile/confidence-level.

**Table 1.** Accuracy comparison between Model 1 and the proposed model.

| Quantile/Confidence-Level | 10% | 20% | 30% | 40% | 50% | 60% | 70% | 80% | 90% | 99% |
|---|---|---|---|---|---|---|---|---|---|---|
| Model 1 | **20%** | **28%** | 30% | 35% | 38% | 43% | 51% | 56% | 60% | 72% |
| proposed model | 12% | 21% | **31%** | **38%** | **46%** | **53%** | **68%** | **74%** | **88%** | **94%** |

*(2) Uncertainty under equal accuracy* (U/A): U/A is obtained by summing the total interval size under the same accuracy. It provides a feasible way to deal with the variable interval size, which is a barrier in comparing the PI and CI methods. Lower uncertainty also means fewer risks of predicting load within a required accuracy level. Comparing Model 2 and proposed model under accuracy of 90%, the results in Figure 12 show that the proposed model can forecast the load with a smaller interval size.

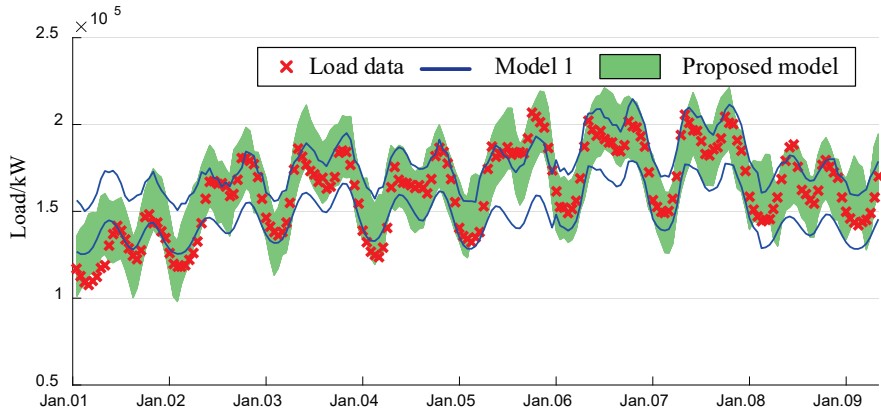

**Figure 11.** Accuracy under the same confidence level (A/CL) comparison between Model 1 and the proposed model under confidence level 50%.

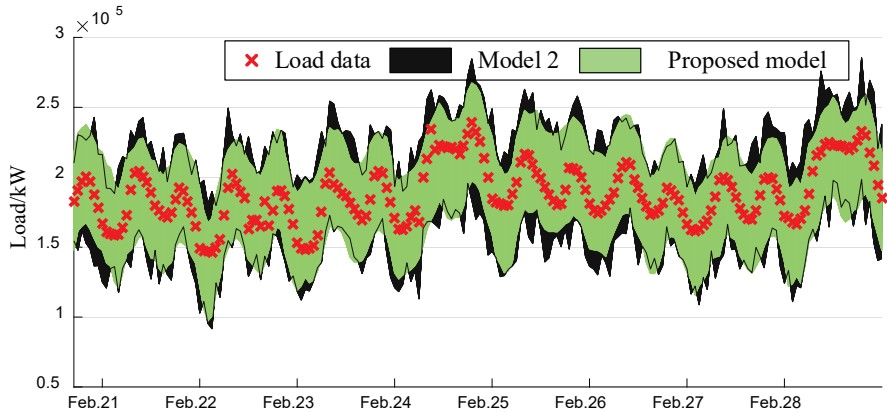

**Figure 12.** Uncertainty under equal accuracy (U/A) comparison between Model 2 and the proposed model under accuracy 90%.

*(3) Accuracy under equal uncertainty* (A/U): A/U is obtained by calculating the accuracy under equal uncertainty ($2z\sigma$). It is designed especially for concise access to distinguish the CI methods, of which the total uncertainty can be easily calculated. As seen in Figure 13, CI methods of Model 3 and proposed model are traced in A/U form. It is clear that the proposed model is more efficient for all given levels of uncertainty. All of the statistical characteristics are described Figure 14 for an intuitive comparison. Influenced by the environmental noise, the raw load data may contain some extreme

values. There is no need to cover all these outliers while forecasting. After eliminating the impact of noise, the fluctuation of forecasting errors obtained from proposed model is smoother. Therefore, lower uncertainty is needed to cover the fluctuation.

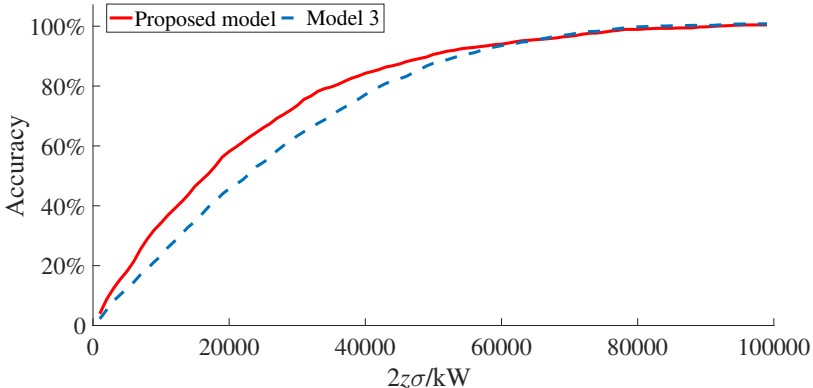

**Figure 13.** Accuracy under equal uncertainty (A/U) comparison between Model 3 and the proposed model.

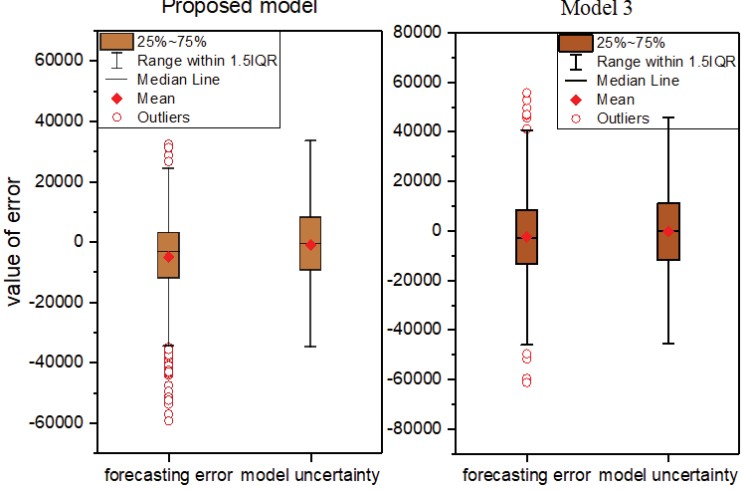

**Figure 14.** Statistical box-plot for forecasting error contrast to uncertainty of the proposed model and Model 3.

Figures 11–14 show graphical comparisons between the proposed model and the contrast models. It is obvious that the proposed model delivers better results in the validations above. To elaborate a little further, the data from four selected stations are devoted to testing all the cases. For a common method to measure the utility of candidate models, an indicator named mean interval proportion (MIP) is introduced as follows:

$$P_\tau = \frac{1}{N} \sum_{i=1}^{N} \frac{U_i^{(\tau)} - L_i^{(\tau)}}{r_i}, \tag{15}$$

where $P_\tau$ is the MIP indicator which means the proportion of interval size in actual load value at coverage $\tau$. $U_i^{(\tau)}$ and $L_i^{(\tau)}$ are respectively the lower and upper thresholds of the interval at coverage $\tau$. The closed intervals formed by the above thresholds represent the least uncertainties at a certain coverage, while the MIPs indicate the maximum tolerance of forecasting errors. Explicitly, the thorough numerical contrast of MIPs among the candidate models can be consulted in Tables 2 and 3.

**Table 2.** Performance comparison of the methods under 50%/90% coverage.

| Method | $\tau = 50\%$ | | | | $\tau = 90\%$ | | | |
|---|---|---|---|---|---|---|---|---|
| | **Model 1** | **Model 2** | **Model 3** | **Proposed** | **Model 1** | **Model 2** | **Model 3** | **Proposed Model** |
| station 1 | 0.067 | 0.072 | 0.077 | **0.058** | 0.268 | 0.159 | 0.154 | **0.114** |
| station 2 | 0.064 | 0.068 | 0.072 | **0.046** | 0.241 | 0.166 | 0.147 | **0.184** |
| station 3 | 0.071 | 0.081 | 0.104 | **0.062** | 0.301 | **0.162** | 0.194 | 0.171 |
| station 4 | 0.066 | 0.076 | 0.081 | **0.049** | 0.255 | 0.184 | 0.139 | **0.108** |

**Table 3.** Performance comparison of the methods under 99%/100% coverage.

| Method | $\tau = 99\%$ | | | | $\tau = 100\%$ | | | |
|---|---|---|---|---|---|---|---|---|
| | **Model 1** | **Model 2** | **Model 3** | **Proposed** | **Model 1** | **Model 2** | **Model 3** | **Proposed Model** |
| station 1 | 0.343 | 0.252 | 0.251 | **0.216** | 0.344 | 0.292 | 0.305 | **0.285** |
| station 2 | 0.321 | 0.267 | 0.260 | **0.202** | 0.324 | **0.289** | 0.342 | 0.301 |
| station 3 | 0.372 | **0.271** | 0.299 | 0.277 | 0.377 | **0.331** | 0.384 | 0.364 |
| station 4 | 0.315 | 0.263 | 0.249 | **0.220** | 0.315 | 0.301 | 0.317 | **0.297** |

## 5. Discussion

The probabilistic forecasting methods discussed in this study tend to provide more detailed information about forecasting output. The proposed model and three state-of-the-art models of quantile, CI and PI methods are simulated for a comprehensive comparison. Validated with numerous data and experiments, prominent results are highlighted in bold. Apparently, the two CI methods hold the steady response. The QRNN method, though having the worst performance at higher coverage, is found to be barely satisfactory at $\tau = 50\%$. The proposed model achieves encouraging results most of the time. One exception is that the performance of the PI method is surprisingly remarkable on station 3, which contains the amount of an outlier due to treacherous weather in its location. Although the CI methods are sensitive to data noise, the proposed integration scheme outperforms other methods in terms of accuracy and can forecast the load with smaller intervals.

## 6. Conclusions

The course of load forecasting is laborious for the sake of various affecting factors in nature. In this paper, an integration scheme of multiple methods is proposed for probabilistic forecasting on short-term electric load demand. To boost the efficiency of probabilistic load forecasting, several improvements are exploited within the integration scheme. Firstly, a cooperation of WPT and correlation analysis is applied to remove data noise. Secondly, the IWELM with a computation reducing switch algorithm is presented for load forecasting. Thirdly, the CIs of forecasting results are produced by the Gaussian process. Finally, the superiority of the proposed model is validated by actual load data from GEFcom2014, and the corresponding criteria are designed for comparison. The results show that the proposed integration scheme is more flexible, reliable and effective than the other available methods.

**Author Contributions:** Z.K. proposed the scheme and drafted the paper. Z.X. prepared the data and contributed analysis methods. Y.C. performed the simulation and modeling. H.L. analyzed the results and modified the paper.

**Funding:** This research was funded by the National Natural Science Foundation of China (NSFC) under grant 61801518 and the Hubei Provincial Natural Science Foundation of China under grant 2017CFB661.

**Conflicts of Interest:** The authors declare no conflict of interest.

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
