# Peer review of "Probabilistic Forecasting of Short-Term Electric Load Demand: An Integration Scheme Based on Correlation Analysis and Improved Weighted Extreme Learning Machine"

_applsci, doi:10.3390/app9204215_

Round 1

Reviewer 1 Report

This paper shows a probabilistic forecasting of short-term electric load demand. Although the proposed method seems better than other state-of-the-art methods, the load forecasting is laborious as there are various affecting factors in nature. I have some questions and minor comments for the authors:

1) Line 230. Tabel instead Table

2) Figure 13, axis - y label should be Accuracy instead accuravy

3) Figure 13. Also, Do this reference to Model 3 or Model 3?

4) Sometimes authors use uppercase for "Model" and other they use lowercase "model"

5) Figure 14. The proposed model has too much outliers. Can you explain them?

6) Are the dataset ranged? When ANN are used, the dataset is used to range, i.e. [0,1]

7) As well, are dataset randomly? This is a practice when ANN are used.

8) k in kW is lowercase, please, the authors should correct this in all figures.

Reviewer 2 Report

Line 100: it is stated that “it is generally accepted that little correlation exists when their absolute values are below 0.3 and the absolute values exceed 0.5 means the series are related”. This statement should be justified with some reference.

Figure 3: Which are the meaning of axes x and y?.

Line 139: “obtained” should be used instead of “obtain”.

Lines 148-162: It should be explained how parameters γk are obtained.

Line 165: Which is the KKT condition?

Lines 165-166: Sentences “Using the KKT conditions. The optimal equation for b is solved as” should be rewritten: maybe they could be combined into an only one.

Figure 5: The text points out (line 180) that Y and b are adjusted, but the left box in the figure states that only Y is adjusted. This should be clarified.

Section 3. Implementation: The meaning and usefulness of applying Gaussian product and convolution should be further explained. It is not clear which are the model outputs, that is to say, how the model provides the confidence interval. This must be clearly explained.

Section 4.1 Data Sets: Data sets should be further explained. Are they only electric demand? Are other data (as temperature, humidity, hour of the day…) used?. If only load data were used it should be explained why other exogenous variables were not taken into account (short term load forecasting usually needs exogenous variables to provide reliable predictions). It sounds strange that load data were obtained from weather stations. This should be clarified. On the other hand it should be explained why so old data were used (data from 2005 and 2006). Why more recent data were not used?

Figures 9 and 10: The meaning of the green strips around data should be further explained in the text. What does the green column to the right side of the figure represent?

Table 1: The data represented in this table must be explained in the text.

The authors should compare their results with those provided in other works to justify that their model actually represents an improvement of knowledge.

Round 2

Reviewer 1 Report

The authors have improved the paper according to the review's comments so this paper deserves to be published.

Reviewer 2 Report

I think the work may be published in present form.

This manuscript is a resubmission of an earlier submission. The following is a list of the peer review reports and author responses from that submission.

Round 1

Reviewer 1 Report

1) Discussion section should be added

2) What are the novelties and contribution here

3) Figure 13. A/U comparison between Model 3 and proposed model

   labels of the x-axis: uncertainty: unit, scale division?

4) Section "2.2. Correlation Analysis"

Pearson correlation coefficient is commonly known. A detailed description unnecessary

Reviewer 2 Report

In this study, an integration scheme composed of correlation analysis and improved weighted extreme learning machine was proposed for probabilistic load forecasting. In this scheme, a novel cooperation of wavelet packet transform and correlation analysis was developed to deal with the data noise. Meanwhile, an improved weighted extreme learning machine with a new switch algorithm was provided to effectively obtain stable forecasting results. The probabilistic forecasting task is then accomplished by generating the confidence intervals with Gaussian process. The proposed integration scheme, tested by actual data from Global Energy Forecasting Competition, is proved to have a better performance in graphic and numerical results than the other available methods.

This subject addressed is within the scope of the journal. However, the manuscript in the present version contains several problems. In addition, this research involves unclear scientific approaches and approximation to solve the addressed problems. Therefore, I cannot accept this manuscripts for the publication. This manuscripts requires the appropriate revisions to justify recommendation for future publication. 

1. It is mentioned that the improved weighted extreme learning machine (IWELM) was used for probabilistic load forecasting. What are the advantages of adopting these data-driven models over others (e.g., MARS, GEP, GRNN, GP, and CCNN etc.) over others in this case? How will this affect the results? More details should be furnished.

2. It is mentioned that the wavelet packet transform and correlation analysis were developed to reduce the data noise. What are the advantages of adopting this the wavelet packet transform and correlation analysis over others in this case? How will this affect the results? More details should be furnished.

3. What are the criteria for evaluating model performance based on probabilistic load forecasting? Only the results of confidence level cannot confirm the forecasting accuracy. Please add other performance criteria and analyze the data.

4. There are some occasional grammatical problems within the text. It may need the attention of someone fluent in English language to enhance the readability.

5. Since all the figures have low resolution printing, the reviewer cannot recognize them clearly. Please revise them with high resolution.

6. The title of manuscript has to be modified. Please consider it.

7. Since the description of used data are not clear, the authors have to explain it clearly.

8. The authors have to add the state-of-the art references in the manuscripts. Relevant researches within worldwide schemes can be found from many journals.

<References>

1. Kong, W., Dong, Z. Y., Jia, Y., Hill, D. J., Xu, Y., Zhang, Y. (2017). Short-term residential load forecasting based on LSTM recurrent neural network. IEEE Transactions on Smart Grid10(1), 841-851.

2. Lang, K., Zhang, M., Yuan, Y., Yue, X. (2018). Short-term load forecasting based on multivariate time series prediction and weighted neural network with random weights and kernels. Cluster Computing, 1-9.

3. Ribeiro, G.T., Mariani, V.C., dos Santos Coelho, L. (2019). Enhanced ensemble structures using wavelet neural networks applied to short-term load forecasting. Engineering Applications of Artificial Intelligence82, 272-281.

4. Yang, Z., Ce, L., Lian, L. (2017). Electricity price forecasting by a hybrid model, combining wavelet transform, ARMA and kernel-based extreme learning machine methods. Applied Energy190, 291-305.

5. Zeng, N., Zhang, H., Liu, W., Liang, J., Alsaadi, F.E. (2017). A switching delayed PSO optimized extreme learning machine for short-term load forecasting. Neurocomputing240, 175-182.

6. Zhang, Y., Liu, K., Qin, L., An, X. (2016). Deterministic and probabilistic interval prediction for short-term wind power generation based on variational mode decomposition and machine learning methods. Energy Conversion and Management112, 208-219.

7. Zheng, H., Yuan, J., Chen, L. (2017). Short-term load forecasting using EMD-LSTM neural networks with a Xgboost algorithm for feature importance evaluation. Energies10(8), 1168.